# Verified Safe Reinforcement Learning for Neural Network Dynamic Models

**Junlin Wu**
Computer Science & Engineering
Washington University in St. Louis
junlin.wu@wustl.edu

**Huan Zhang**
Electrical & Computer Engineering
University of Illinois Urbana-Champaign
huan@huan-zhang.com

**Yevgeniy Vorobeychik**
Computer Science & Engineering
Washington University in St. Louis
yvorobeychik@wustl.edu

## Abstract

Learning reliably safe autonomous control is one of the core problems in trust-worthy autonomy. However, training a controller that can be formally verified to be safe remains a major challenge. We introduce a novel approach for learning verified safe control policies in nonlinear neural dynamical systems while maximizing overall performance. Our approach aims to achieve safety in the sense of finite-horizon reachability proofs, and is comprised of three key parts. The first is a novel curriculum learning scheme that iteratively increases the verified safe horizon. The second leverages the iterative nature of gradient-based learning to leverage incremental verification, reusing information from prior verification runs. Finally, we learn multiple verified initial-state-dependent controllers, an idea that is especially valuable for more complex domains where learning a single universal verified safe controller is extremely challenging. Our experiments on five safe control problems demonstrate that our trained controllers can achieve verified safety over horizons that are as much as an order of magnitude longer than state-of-the-art baselines, while maintaining high reward, as well as a perfect safety record over entire episodes. Our code is available at https://github.com/jlwu002/VSRL.

## 1 Introduction

The ability to synthesize safe control policies is one of the core challenges in autonomous systems. This problem has been explored from numerous directions across multiple disciplines, including control theory and AI [Achiam et al., 2017, Dawson et al., 2022]. While considerable progress has been made, particularly when dynamics are linear [Wabersich and Zeilinger, 2018], the ability to synthesize controllers that can be successfully *verified* to be safe while maintaining high performance in nonlinear dynamical systems remains a major open problem. Indeed, even the subproblem of safety verification in nonlinear systems is viewed in itself as a major challenge and is an active area of research, particularly for neural network controllers [Bastani et al., 2018, Ivanov et al., 2019, Wei and Liu, 2022]. State-of-the-art approaches for safe control synthesis, including most that leverage reinforcement learning [Gu et al., 2022], typically only offer empirical evaluation of safety, and rely on safety proofs that hold either asymptotically (rather than for concrete problems) [Xiong et al., 2024] or under idealized assumptions which do not hold in practice [Berkenkamp et al., 2017].

Two common properties are typically leveraged in safety verification: *forward invariance* and *reachability*. The former aims to identify a set of starting subsets of safe states under which one-step

(forward) dynamics remain in this (forward invariant) set. The latter computes the set of states that can possibly be reached after $K$ steps of the dynamics for a given control policy, and checks whether it intersects with the unsafe set. Approaches for synthesizing (including those that do so using learning) safe policies almost exclusively aim to achieve verified safety through forward invariance. However, this has proved extremely challenging to employ beyond the simplest dynamics.

We propose the first (to our knowledge) approach for learning $K$-step verified safe neural network controllers that also aim to maximize efficiency in systems with neural dynamics. While neural dynamics are clearly not universal, they can capture or effectively approximate a broad range of practical dynamical systems [Nagabandi et al., 2018], and have consequently been the focus of much prior work in safe control and verification [Dai et al., 2021]. For example, consider the scenario of a drone navigating through a series of obstacles to reach a designated goal, requiring $K = 50$ steps to safely maneuver through the obstacles. We aim to train a controller that can reach the goal as fast as possible, while guaranteeing safety for the initial $50$ steps, ensuring 1) the drone does not collide with any obstacles and 2) its angle remains within a predefined safe range.

Our approach combines deep reinforcement learning with state-of-the-art differentiable tools for efficient reachability bound computation, and contains two key novel ingredients. The first is a novel curriculum learning scheme for learning a verified safe controller. This scheme takes advantage of the structure of the $K$-reachability problem at the root of our safety verification by creating a curriculum sequence with respect to increasing $K$. An important insight that is specific to the verification setting is that verification must work not merely for a fixed $K$, but for all steps prior, an issue we address by memorizing subsets of states who either violate, or nearly violate, safety throughout the entire $K$-step curriculum learning process. Additionally, to maintain *both* strong empirical and verified performance, we propose a novel loss function that integrates overall reward, as well as *both* traditional (empirical) safety loss along with the $K$-reachability bound. Our second innovation is to learn a *collection* of controllers that depend on the initial state, in contrast to typical approaches that focus on learning a single "universal" controller. The ability to allow for learning multiple controllers makes the verified learning problem considerably easier, as we can "save" controllers that work on a subset of initial states, and simply try learning a new controller for the rest, guaranteeing incremental improvement through the learning process. We further improve performance through incremental verification, which leverages information obtained in previous learning iterations.

We evaluate the proposed approach in five control settings. The first two are lane following and obstacle avoidance, both pertaining to autonomous driving. The last three involve drone control with obstacle avoidance. Two of these consider fixed obstacles, while the third aims to avoid even moving obstacles (with known dynamics). We show that the proposed approach outperforms five state-of-the-art safe control baselines in the ability to achieve verified safety without significantly compromising overall reward (efficiency). In particular, our approach learns controllers that can verify $K$-step safety for $K$ up to an order of magnitude larger than the prior art and maintains a perfect safety record for $K$ far above what we verify, something no baseline can achieve.

In summary, we make the following contributions:

1. A framework for safe optimal control that combines both finite-horizon verified (worst-case) and empirical (average-case) safety constraints.
2. A novel curriculum learning approach that leverages memorization, forward reachability analysis, and differentiable reachability overapproximation for efficiently learning verified safe policies.
3. An approach for learning a *collection* of control policies that depend on the initial state which enables significant improvements in verified safety horizon over large initial state sets $S_0$.
4. An incremental verification approach that leverages small changes in gradient-based learning to improve verification efficiency during learning.
5. An extensive experimental evaluation that demonstrates the efficacy of the proposed approach in comparison with five state-of-the-art safe RL baselines.

**Related Work:** Safe reinforcement learning has been extensively studied through the lens of constrained Markov decision process (CMDP)-based approaches, which represent cost functions as constraints and aim to maximize reward while bounding cost, using approaches such as Lagrangian and penalty methods, and constrained policy optimization [Achiam et al., 2017, Stooke et al., 2020, Ma et al., 2022, Jayant and Bhatnagar, 2022, Yu et al., 2022, So and Fan, 2023, Ganai et al., 2024].

An alternative control-theoretic perspective aims to ensure stability or safety using Lyapunov and control barrier functions. For example, Dawson et al. [2022] used a learning-based approach to find robust control Lyapunov barrier functions; Chow et al. [2018] constructed Lyapunov functions to solve CMDPs; [Wang et al., 2023a] proposed soft barrier functions for unknown and stochastic environments; and Alshiekh et al. [2018] created safety shielding for safe RL agents. These approaches, however, provide no practical formal safety guarantee for neural network controllers. In addition, some work on provably safe RL focuses on the probabilistic setting [Berkenkamp et al., 2017, Jansen et al., 2020, Xiong et al., 2024] and required statistical assumptions, whereas our work aims for strict deterministic safety guarantees over a finite horizon.

Among existing works focusing on safe RL with formal guarantees, Fulton and Platzer [2018] apply a theorem prover for differential dynamic logic to guarantee safety during runtime. Noren et al. [2021] and Wei et al. [2022] consider forward safety invariance for systems with uncertainty. Kochdumper et al. [2023] propose to project actions to safe subspace using zonotope abstraction and mixed-integer programming (MIP). However, these approaches do not readily apply to neural network controllers. For systems involving neural networks, Wei and Liu [2022] applied integer programming formulation for neural networks to solve an MIP problem to find safe control actions satisfying forward invariance; Bastani et al. [2018] extracted decision-tree-based policies for RL to reduce verification complexity; and Ivanov et al. [2019] used hybrid system verification tools to model deep neural networks. Our work differs from these and similar approaches because we consider forward reachability guarantees for neural network controllers in neural nonlinear systems.

We make extensive use of neural network verification tools. Early work in this vein used SMT [Katz et al., 2017, Huang et al., 2017] or MIP-based [Tjeng et al., 2019] approaches to solve this problem, but their scalability is extremely limited. Significant progress has been made in developing techniques to formally verify the properties of large neural networks through overapproximation, such as bound propagation [Zhang et al., 2018, Gowal et al., 2018, Xu et al., 2021], optimization [Qin et al., 2019, Dvijotham et al., 2018, 2020], and abstract interpretation [Gehr et al., 2018, Singh et al., 2019, Katz et al., 2019, Lopez et al., 2023]. Recently, most verifiers have adopted branch-and-bound based approaches to further enhance their performance [Wang et al., 2021, Kouvaros and Lomuscio, 2021, Ferrari et al., 2022, Zhang et al., 2022]. Our approach makes use of differentiable overapproximation methods known collectively as $\alpha,\beta$-CROWN [Wang et al., 2021, Zhang et al., 2022] (implemented with the auto_LiRPA package), and takes advantage of the particular structure of these verification approaches in applying incremental verification to significantly speed up safe controller learning.

## 2 Preliminaries

**Constrained Markov Decision Process (CMDP):** We consider a deterministic Constrained Markov Decision Process (CMDP) defined by the tuple $(\mathcal{S}, \mathcal{A}, F, R, \gamma, C_1, C_2, \ldots, C_m, d_1, d_2, \ldots, d_m)$, where: $\mathcal{S}$ is a set of states, $\mathcal{A}$ is a set of actions, $F : \mathcal{S} \times \mathcal{A} \to \mathcal{S}$ is the deterministic state transition function, $R : \mathcal{S} \times \mathcal{A} \to \mathbb{R}$ is the reward function, $C_i : \mathcal{S} \times \mathcal{A} \to \mathbb{R}$ is the cost function for the $i$-th constraint, $d_i$ is the cost limit for the $i$-th constraint, and $\gamma \in [0, 1)$ is the discount factor. A policy $\pi : \mathcal{S} \to \mathcal{A}$ is a mapping from states to actions. A trajectory is a sequence of states and actions generated by following a policy $\pi$ from some initial state $s_0 \in \mathcal{S}_0 \subseteq \mathcal{S}$, which can be represented as a sequence $\tau = (s_0, a_0, s_1, a_1, s_2, a_2, \ldots)$ where $s_t \in \mathcal{S}$, $a_t = \pi(s_t)$ for all $t$, $s_{t+1} = F(s_t, a_t)$, a reward $r_t = R(s_t, a_t)$ and a cost $c_t = \sum_{i \in [m]} C_i(s_t, a_t)$ are received after each action.

We denote $\pi_\theta$ as the policy that is parameterized by the parameter $\theta$. A common goal for CMDP is to learn a policy $\pi_\theta$ that maximizes a discounted sum of rewards $\mathcal{J}(\pi_\theta)$ while ensuring that expected discounted costs $\mathcal{J}_{C_i}(\pi_\theta)$ do not exceed the cost limit $d_i, \forall i \in [m]$. Formally, CMDP is to solve the below optimization problem:

$$\max_\theta \mathcal{J}(\pi_\theta) \quad \text{s.t. } \mathcal{J}_{C_i}(\pi_\theta) \leq d_i, \forall i \in [m], \tag{1}$$

where $\mathcal{J}(\pi_\theta) = \mathbb{E}_{\tau \sim \pi} \left[ \sum_{t=0}^{\infty} \gamma^t R(s_t, a_t) \right]$ and $\mathcal{J}_{C_i}(\pi_\theta) = \mathbb{E}_{\tau \sim \pi} \left[ \sum_{t=0}^{\infty} \gamma^t C_i(s_t, a_t) \right]$.

**Verified Safe CMDP:** We define the state space as the union of predefined safe and unsafe states, denoted as $\mathcal{S} = \mathcal{S}_{\text{safe}} \cup \mathcal{S}_{\text{unsafe}}$. We assume that the transition function $F$ is represented by a ReLU neural network, and is known for verification purposes. This assumption is very general, as many known dynamical systems can be represented exactly or approximately using ReLU neural networks [Gillespie et al., 2018, Pfrommer et al., 2021, Dai et al., 2021, Liu et al., 2024]. Our

objective is to train a controller that not only satisfies safety constraints empirically at decision time, but also ensures verified safety for the first $K$ steps.

Formally, we aim to solve the following optimization problem:

$$\max_{\theta} \mathcal{J}(\pi_\theta) \tag{2a}$$

$$\text{s.t. } \mathcal{J}_{C_i}(\pi_\theta) \leq d_i, \quad \forall i \in [m] \quad \text{(empirically satisfied)} \tag{2b}$$

$$s_t \in \mathcal{S}_{\text{safe}}, \quad \forall t \in [K] \quad \text{(mathematically verified)} \tag{2c}$$

$$s_{t+1} = F(s_t, a_t), a_t = \pi_\theta(s_t), s_0 \in \mathcal{S}_0 \subseteq \mathcal{S}_{\text{safe}} \tag{2d}$$

In particular, we aim to solve (2) for high values of $K$ and large sets of verified safe initial states $\mathcal{S}_0$, while preserving a high objective value. Note that for a given controller, (2c) can also be interpreted as a set of forward reachability verification problems. However, our interest here extends beyond mere verification; we aim to *train (synthesize) a controller that can be efficiently verified for safety*. For simplicity, we restrict attention to $d_i = 0$ for all $i$; however, our approach can be directly applied to arbitrary values of $d_i$.

In this work, we primarily utilize the $\alpha,\beta$-CROWN toolbox [Xu et al., 2021, Wang et al., 2021] for neural network (NN) verification; however, our training framework is general and, in principle, can work with any *differentiable* verification technique. Let $F^{k,\pi_\theta}$ denote the $k$-step forward function (iterative composition of $F$) under policy $\pi_\theta$. For example, $s_1 = F^{1,\pi_\theta}(s) = F(s, \pi_\theta(s))$, $F^{2,\pi_\theta}(s) = F(s_1, \pi_\theta(s_1))$, and so on. Correspondingly, we represent the $k$-step forward reachable regions returned by the NN verifier for an initial state set $S$ as $F_{\text{Bound}}^{k,\pi_\theta}(S)$, which is typically represented as a box.

## 3 Approach

The problem of learning verified safe control over a target horizon $K$ entails three key technical challenges. The first is that as $K$ grows, the differentiable overapproximation techniques for reachability verification become looser, making it difficult to verify $K$ beyond very small horizons. Second, while control policies $\pi$ depend on state, it is difficult to find a single *universal* controller that can achieve verified safety for each starting state in $\mathcal{S}_0$. Our approach addresses these challenges through three technical advances: 1) curriculum learning with memorization and 2) incremental verification, which enable learning verified safe controllers over longer horizons $K$, and 3) iterative learning a collection of controllers customized for subsets of $\mathcal{S}_0$, which addresses the third challenge above.

**Curriculum Learning with Memorization:** Curriculum learning is an iterative training strategy where the difficulty of the task increases as training progresses [Bengio et al., 2009, Wu and Vorobeychik, 2022]. At a high level, for a problem targeting $K$-step verified safety, training can be divided into $K$ phases, with each phase $k$ aiming to achieve verified safety at the corresponding $k$-th forward step. In the $k$-th phase, we conduct formal verification against the $k$-th step safety, filter out regions that cannot be verified, and use them for further training. As $k$ increases, the task difficulty also increases, mainly due to the forward NN $F^{k,\pi_\theta}$ becoming deeper. For a deeper NN and a fixed branching budget, the output bounds become looser [Wang et al., 2021], increasing the likelihood of intersections with unsafe regions. However, the ability to verify safety in prior steps enables us to tailor a controller that closely aligns with the fixed NN dynamics, thereby achieving tighter bounds. This process captures the essence of curriculum learning.

Nevertheless, our approach deviates from traditional curriculum learning in a way that is quite consequential for our setting: we aim to ensure that a controller is verified as safe not only for the $k$-th step but also maintains safety for all prior steps. Consequently, during our curriculum learning process, we store states that are close to being unsafe in each phase in a buffer, effectively memorizing information about regions that potentially violate safety. These states, along with the unverified states at the current phase, are then incorporated into the training process, helping to ensure safety across the entire $K$-step horizon. Our curriculum training framework is detailed in Algorithm 1.

The process begins by initializing the policy $\pi_\theta$ with a pre-trained policy using safe RL algorithms (Line 3). We then split the initial region $\mathcal{S}_0$ into a grid $\mathcal{G}_0$ (Line 4). We assume $\mathcal{S}_0$ is a $m$-dimensional box centered at $s_c \in \mathbb{R}^m$ with a radius $r \in \mathbb{R}^m$, i.e., $\mathcal{S}_0 = [s_c - r, s_c + r]$. We prioritize splitting the dimensions that are directly implicated in safety constraints, thereby taking advantage of the

---

**Algorithm 1** Curriculum Learning with Memorization

---

1: **Input:** target safety horizon $K$, initial region $\mathcal{S}_0$, unsafe region $\mathcal{S}_{\text{unsafe}}$, max attempt $n_{\max}$
2: **Output:** controller $\pi_\theta$
3: Initialize $\pi_\theta$ with pre-trained policy, buffer $B = \{\}$
4: Split initial region $\mathcal{S}_0$ into grid $\mathcal{G}_0$
5: **for** $k = 1, 2, \ldots, K$ **do**
6:     $n_{\text{train}} \leftarrow 0$
7:     $S_{uc} \leftarrow F_{\text{Bound}}^{k,\pi_\theta}(\mathcal{G}_0) \cap \mathcal{S}_{\text{unsafe}}$ // optionally use Branch-and-Bound to refine $\mathcal{G}_0$
8:     **while** $n_{\text{train}} < n_{\max}$ and $S_{uc} \neq \emptyset$ **do**
9:         Safe RL training with loss function $\mathcal{L}(x) = \mathcal{L}_{\text{SafeRL}}(x) + \lambda \mathcal{L}_{\text{Bound}}(S_{uc} \cup B)$
10:        $n_{\text{train}} \leftarrow n_{\text{train}} + 1$
11:        $S_{uc} \leftarrow F_{\text{Bound}}^{k,\pi_\theta}(\mathcal{G}_0) \cap \mathcal{S}_{\text{unsafe}}$
12:     **end while**
13:     Filter regions $S_k \subseteq \mathcal{G}_0$ such that $\text{dist}(F_{\text{Bound}}^{k,\pi_\theta}(S_k), \mathcal{S}_{\text{unsafe}}) < \epsilon$, store $(S_k, k)$ in buffer $B$
14: **end for**

---

typical structure of safety constraints that only pertain to a small subset of state variables. For instance, in drone control for obstacle avoidance, we prioritize splitting the location and angle axes. Next, we design a cost function $C_R$ for regions where $C_R(S) = 0$ if $S \cap \mathcal{S}_{\text{unsafe}} = \emptyset$, and $C_R(S) > 0$ otherwise. A positive $C_R$ means region $S$ intersects with $\mathcal{S}_{\text{unsafe}}$, while $C_R = 0$ indicates $S$ is safe. For example, if the task is to avoid the region $[a, b]$, and the output bounds are given by $x_B = [x_{lb}, x_{ub}]$, we can define $C_R(x_B) = \max(x_{ub} - a, 0) \cdot \max(b - x_{lb}, 0)$. We then calculate the gradient $\partial C_R(F_{\text{Bound}}^{t,\pi_\theta}(\mathcal{S}_0))/\partial r$ for a chosen value of $t$ and proceed to split along the dimensions with the largest gradient values, as a larger gradient indicates a higher likelihood of reducing the cost $C_R$. We continue this process, keeping the total number of grid splits within a predetermined budget, and stop splitting once the budget is reached.

For each training phase $k$, we monitor the training rounds ($n_{\text{train}}$) as well as the $k$-step forward reachable regions returned by the verifier that are identified as unsafe ($S_{uc}$). Each phase is conducted for a maximum of $n_{\max}$ rounds or until verified $k$-step safety is achieved, that is, when $S_{uc} = \emptyset$ (Line 8). At the end of each training phase, we also filter out regions $S_k \subseteq \mathcal{G}_0$ where $F_{\text{Bound}}^{k,\pi_\theta}(S_k)$ are within $\epsilon$ distance to the unsafe regions. These regions are then stored in the buffer $B$ (Line 13). We include these critical regions in the training set for each reinforcement learning update to enhance verified safety across the entire horizon. During this process, we optionally use the Branch-and-Bound algorithm [Everett et al., 2020, Wang et al., 2021] to refine $\mathcal{G}_0$ up to a predetermined branching limit, which helps achieve tighter bounds.

For each RL update, we use a loss function that integrates the standard safe RL loss with a $k$-phase loss for bounds (Line 9), where

$$\mathcal{L}(x) = \mathcal{L}_{\text{SafeRL}}(x) + \lambda \mathcal{L}_{\text{Bound}}(S_{uc} \cup B) \tag{3a}$$

$$\mathcal{L}_{\text{Bound}}(S_{uc} \cup B) = C_R(F_{\text{Bound}}^{k,\pi_\theta}(S_{uc})) + \sum_{(S_i, i) \in B} C_R(F_{\text{Bound}}^{i,\pi_\theta}(S_i)). \tag{3b}$$

Here, $\mathcal{L}_{\text{SafeRL}}$ is the standard safety RL loss, and $\mathcal{L}_{\text{Bound}}$ denotes the loss that incentivizes ensuring the output bounds returned by the verifier remain within the safe region. If both $S_{uc}$ is $k$-step safe and $\forall (S_i, i) \in B$, $S_i$ is $i$-step safe, then $\mathcal{L}_{\text{Bound}}(S_{uc} \cup B) = 0$, otherwise, $\mathcal{L}_{\text{Bound}}(S_{uc} \cup B) > 0$. In practice, we clip $\mathcal{L}_{\text{Bound}}(S_{uc} \cup B)$ to ensure it remains within a reasonable range for training stability. The regularization parameter $\lambda$ is calculated based on the magnitude of $\mathcal{L}_{\text{SafeRL}}$ and $\mathcal{L}_{\text{Bound}}$, with $\lambda = \min(\lambda_{\max}, a_r \cdot \mathcal{L}_{\text{SafeRL}}/\mathcal{L}_{\text{Bound}})$, where $\lambda_{\max}$ and $a_r$ are hyperparamters. This approach helps maintain the effectiveness of bound training, especially when $\mathcal{L}_{\text{Bound}}$ is small. Furthermore, we cluster elements in $B$ into categories so that we do not need to construct a computational graph for all $i < k$. Specifically, we merge all $S_i$ for $i_1 \leq i \leq i_2$ into the $i_2$ category, meaning the elements in $B$ are now $(\cup_{i_1 \leq i \leq i_2} S_i, i_1, i_2)$ instead of $(S_i, i)$.

It is important to note that while our training scheme targets $K$-step verified safety, the policy returned by Algorithm 1 does not necessarily guarantee it. We address this issue by learning initial-state-dependent controllers as described below. Furthermore, the computation of $\mathcal{L}_{\text{Bound}}$ is computationally intensive. Its backpropagation requires constructing computational graphs for the $k$-th step forward

NN $F_{\text{Bound}}^{k,\pi_\theta}$, as well as for all $i$-th step forward NNs corresponding to each $(S_i, i) \in B$. These NNs become increasingly deep as $k$ grows, causing the computational graphs to consume memory beyond the typical GPU memory limits. We will address this next.

**Incremental Verification:** Above we discussed the challenge presented by the backpropagation of $\mathcal{L}_{\text{bound}}$, which is GPU-memory intensive and does not scale efficiently as the target $K$-step horizon increases. To mitigate these issues, we propose the use of incremental verification to enhance computational efficiency and reduce memory consumption. While incremental verification is well-explored in the verification literature Wang et al. [2023b], Althoff [2015], to our knowledge, we are the first to apply it in *training* provably safe controllers.

At a high level, to calculate the reachable region for a $k_{\text{target}}$ step, we decompose the verification into multiple phases. We begin by splitting the $k_{\text{target}}$ horizon into intervals defined by $0 < k_1 < k_2 < \cdots < k_n = k_{\text{target}}$. We first calculate the reachability region for the $k_i$ step and then use its output bounds as input to calculate the reachable region for the $k_{i+1}$ step. This approach ensures that the computational graph is only built for the $(k_{i+1} - k_i)$ step horizon when using $\alpha,\beta$-CROWN.

Unlike traditional incremental verification, which typically calculates the reachable region from $k$ to $k+1$, we incrementally verify and backpropagate several steps ahead in a single training iteration (i.e., from $k_i$ to $k_{i+1}$, where $k_{i+1} - k_i > 1$). This generalized version of incremental verification is essential for training, as it significantly accelerates the process and reduces the likelihood of becoming trapped in "local optima," where inertia from the policy obtained for $k$ prevents successful verification for $k+1$ (e.g., due to proximity to the unsafe region with velocity directed toward it).

For the bounds used in neural network training, we effectively build the computational graph and perform backpropagation using a neural network sized for $(k_n - k_{n-1})$ steps' reachability, which is independent of $k_{\text{target}}$. This significantly reduces GPU memory usage. Since $F^{k,\pi_\theta}$ is an iterative composition of $F$ under the same policy $\pi_\theta$, the bound for $k_{n-1}$ steps tends to be tight. Moreover, when training $\pi_\theta$ to tighten these bounds, the overall bound for the entire $k_{\text{target}}$ horizon becomes increasingly tight.

**Initial-State-Dependent Controller:** While curriculum learning above includes verification steps, it does not guarantee verified safety for the controller over the entire $K$-step horizon. In this section, we propose using an initial-state-dependent controller to address this issue. For example, in a vehicle avoidance scenario, different initial conditions, such as varying speeds and positions, may correspond to different control strategies. We introduce a mapping function $h : \mathcal{S}_0 \rightarrow \Theta$, which maps each initial state $s_0 \in \mathcal{S}$ to a specific policy $\pi_{h(s_0)}$. The underlying idea is that training a verifiable safe policy $\pi_\theta$ over the entire set of initial states $\mathcal{S}_0$ is inherently challenging. However, by mapping each initial state to a specific set of parameters, we can significantly enhance the expressivity of the policy. This approach is particularly effective in addressing and eliminating corner cases in unverifiable regions.

At a high level, the mapping and parameter set $\Theta$ are obtained by first performing comprehensive verification for the controller output from Algorithm 1 over the entire $K$-step horizon. We then filter unverified regions, cluster them, and fine-tune the controller parameters $\theta$ for each cluster. We store these fine-tuned parameters in the parameter set $\Theta$. This iterative refinement process continues until for every $s_0 \in \mathcal{S}_0$, there exists a $\theta \in \Theta$ such that $\pi_\theta$ is verified safe for the entire $K$-step horizon. The detailed algorithm is presented in Algorithm 2.

The algorithm starts with verifying the policy $\pi_\theta$ obtained from Algorithm 1. The function $\text{VERIFYSAFE}(\pi_\theta, \mathcal{S}_0, K)$ (Line 4) performs verification of policy $\pi_\theta$ for initial states $\mathcal{S}_0$ for the entire horizon $K$. This verification process identifies and categorizes regions into verified safe areas, $S_{\text{safe}}^V$, and areas identified as unsafe, $S_{\text{unsafe}}^V$. Notably, the union of these regions covers all initial states, meaning $S_{\text{safe}}^V \cup S_{\text{unsafe}}^V = \mathcal{S}_0$. After verifying that any state $s_0 \in S_{\text{safe}}^V$ is guaranteed to be safe under policy $\pi_\theta$, we record $(S_{\text{unsafe}}^V, \pi_\theta)$ in the mapping dictionary $H$ (Line 5).

Next, we address the unsafe regions $S_{\text{unsafe}}^V$ that lack a corresponding verified safe controller. We first cluster them based on the type of safety violation (Line 7). The reason for clustering is that regions with similar safety violations are more likely to be effectively verified safe by the same controller. For instance, in a scenario involving navigation around two obstacles, we could potentially identify up to three clusters: the first corresponding to grids that can lead to collisions with obstacle 1, the second includes grids associated with collisions with obstacle 2, and the third is the set of grids that may lead to collisions with both. Given the finite number of safety constraints, the number of possible clusters is also finite. Although the theoretical maximum number of clusters grows exponentially

---
**Algorithm 2** Initial-State-Dependent Controller
---
1: **Input:** target safety horizon $K$, policy $\pi_\theta$
2: **Output:** mapping dictionary $H$, which includes the mapping $h$ and parameter set $\Theta$
3: Initialize $H = \{\}$
4: $(S^V_{\text{safe}}, S^V_{\text{unsafe}}) \leftarrow \text{VERIFYSAFETY}(\pi_\theta, \mathcal{S}_0, K)$
5: Store $(S^V_{\text{safe}}, \theta)$ in mapping dictionary $H$
6: **while** $S^V_{\text{unsafe}} \neq \emptyset$ **do**
7: $\quad \{S_1, S_2, \ldots, S_I\} \leftarrow \text{CLUSTERREGION}(S^V_{\text{unsafe}})$ // cluster based on safety violation
8: $\quad$ **for** $i = 1, 2, \ldots, I$ **do**
9: $\quad\quad \pi_{\theta'} \leftarrow \text{TRAINPOLICY}(\pi_\theta, S_i, K)$
10: $\quad\quad (S^V_{\text{safe},i}, S^V_{\text{unsafe},i}) \leftarrow \text{VERIFYSAFETY}(\pi_{\theta'}, S_i, K)$
11: $\quad\quad$ Store $(S^V_{\text{safe},i}, \theta')$ in mapping dictionary $H$
12: $\quad$ **end for**
13: $\quad S^V_{\text{unsafe}} \leftarrow \bigcup_i S^V_{\text{unsafe},i}$
14: **end while**
---

with the number of safety constraints, in practice, this number is significantly smaller. This is due to the fact that the controller, being pretrained, is less likely to violate multiple or all constraints simultaneously. We then fine-tune the controller for the initial states in each cluster using Algorithm 1. This fine-tuning process is typically fast, as the initial policy is already well-trained. We store each initial state region and its corresponding verified safe policy in the mapping dictionary $H$. This clustering and fine-tuning process continues until a verified safe policy exists for every $s_0 \in \mathcal{S}_0$.

At decision time, given an initial state $s_0$, we first identify the pair $(S^V_{\text{safe}}, \pi_\theta)$ in the mapping dictionary $H$ where $s_0 \in S^V_{\text{safe}}$, then use the corresponding verified safe controller $\pi_\theta$. Note that the soundness of the algorithm directly follows from our use of the sound verification tool $\alpha,\beta$-CROWN.

# 4 Experiments

## 4.1 Experiment Setup

We evaluate our proposed approach in five control settings: Lane Following, Vehicle Avoidance, 2D Quadrotor (with both fixed and moving obstacles), and 3D Quadrotor [Kong et al., 2015, Dai et al., 2021]. The dynamics of these environments are approximated using NN with ReLU activations. We use a continuous action space for those discrete-time systems. In each experiment, we specify the initial region $\mathcal{S}_0$ for which we wish to achieve verified safety. We then aim to achieve the maximum $K$ for which safety can be verified. We evaluate the approaches using four metrics: 1) *Verified-K*: the percentage of regions in $\mathcal{S}_0$ that can be verified for safety over $K$ steps; 2) *Verified-Max*: the maximum number of steps for which all states in $\mathcal{S}_0$ can be verified as safe; 3) *Emp-k*: the percentage of regions in $\mathcal{S}_0$ that are empirically safe for $k$ steps, obtained by sampling $10^7$ datapoints from the initial state $\mathcal{S}_0$. This is evaluated for both $k = K$ (the number of steps we are able to verify safety for) and $k = T$ (total episode length); 4) *Avg Reward*: the average reward over 10 episodes, with both mean and standard deviations reported. Note that the average reward is computed over the entire episode horizon for each environment, independently of the verification horizon, as in conventional reinforcement learning.

We compare the proposed *verified safe RL (VSRL)* approach to six baselines: 1) PPO-Lag, which utilizes constrained PPO with the standard Lagrangian penalty [Achiam et al., 2017]; 2) PPO-PID, which employs constrained PPO with PID Lagrangian methods [Stooke et al., 2020]; 3) CAP, which adopts model-based safe RL with an adaptive penalty [Ma et al., 2022]; 4) MBPPO, which applies model-based safe RL with constrained PPO [Jayant and Bhatnagar, 2022]; 5) CBF-RL, which is a Control Barrier Function (CBF)-based safe reinforcement learning approach [Emam et al., 2022]; and 6) RESPO, which implements safe RL using iterative reachability estimation [Ganai et al., 2024].

Next, we describe the four autonomous system environments in which we run our experiments. Further experimental setup details are provided in Appendix A.2.

Table 1: Results for verified safety, empirical safety and average reward. The percentage results are truncated instead of rounded, to prevent missing unsafe violations.

| | Verified-80(↑) | Verified-Max(↑) | Emp-80(↑) | Emp-500(↑) | Avg Reward(↑) |
|---|---|---|---|---|---|
| Lane Following | | | | | |
| PPO-Lag | 98.6 | 7 | 99.9 | 99.9 | $326 \pm 6$ |
| PPO-PID | 88.5 | 8 | 99.9 | 99.9 | $327 \pm 6$ |
| CAP | 99.5 | 7 | 99.9 | 99.9 | $357 \pm 4$ |
| MBPPO | 99.7 | 8 | 99.9 | 99.9 | $382 \pm 5$ |
| CBF-RL | 98.7 | 7 | 99.9 | 99.9 | $331 \pm 7$ |
| RESPO | 99.8 | 7 | 99.9 | 99.9 | $\mathbf{383 \pm 7}$ |
| VSRL | **100.0** | **80** | **100.0** | **100.0** | $214 \pm 5$ |

| | Verified-50(↑) | Verified-Max(↑) | Emp-50(↑) | Emp-500(↑) | Avg Reward(↑) |
|---|---|---|---|---|---|
| Vehicle Avoidance (Moving Obstacles) | | | | | |
| PPO-Lag | 72.8 | 6 | 87.8 | 87.8 | $303 \pm 12$ |
| PPO-PID | 72.0 | 6 | 89.4 | 89.4 | $287 \pm 22$ |
| CAP | 73.3 | 13 | 89.5 | 89.5 | $393 \pm 35$ |
| MBPPO | 82.6 | 6 | 94.2 | 94.2 | $375 \pm 10$ |
| CBF-RL | 73.0 | 6 | 89.3 | 89.3 | $301 \pm 15$ |
| RESPO | 74.5 | 9 | 89.6 | 89.6 | $391 \pm 20$ |
| VSRL | **100.0** | **50** | **100.0** | **100.0** | $\mathbf{401 \pm 4}$ |

| | Verified-50(↑) | Verified-Max(↑) | Emp-50(↑) | Emp-500(↑) | Avg Reward(↑) |
|---|---|---|---|---|---|
| 2D Quadrotor (Fixed Obstacles) | | | | | |
| PPO-Lag | 0.0 | 5 | 83.4 | 83.4 | $405 \pm 30$ |
| PPO-PID | 0.0 | 4 | 99.3 | 97.5 | $\mathbf{411 \pm 25}$ |
| CAP | 0.0 | 3 | 99.5 | 99.5 | $393 \pm 12$ |
| MBPPO | 58.9 | 9 | 99.9 | 84.5 | $399 \pm 11$ |
| CBF-RL | 0.0 | 5 | 89.9 | 89.7 | $408 \pm 17$ |
| RESPO | 60.4 | 14 | 99.9 | 99.9 | $339 \pm 19$ |
| VSRL | **100.0** | **50** | **100.0** | **100.0** | $401 \pm 20$ |

| | Verified-50(↑) | Verified-Max(↑) | Emp-50(↑) | Emp-500(↑) | Avg Reward(↑) |
|---|---|---|---|---|---|
| 2D Quadrotor (Moving Obstacles) | | | | | |
| PPO-Lag | 0.0 | 3 | 99.7 | 99.7 | $371 \pm 7$ |
| PPO-PID | 0.0 | 2 | 99.7 | 99.7 | $371 \pm 5$ |
| CAP | 57.1 | 8 | 99.2 | 99.2 | $362 \pm 3$ |
| MBPPO | 0.0 | 4 | 99.3 | 99.3 | $\mathbf{374 \pm 6}$ |
| CBF-RL | 0.0 | 4 | 99.3 | 99.3 | $369 \pm 6$ |
| RESPO | 0.0 | 6 | 99.1 | 99.1 | $373 \pm 6$ |
| VSRL | **100.0** | **50** | **100.0** | **100.0** | $364 \pm 4$ |

| | Verified-15(↑) | Verified-Max(↑) | Emp-15(↑) | Emp-500(↑) | Avg Reward(↑) |
|---|---|---|---|---|---|
| 3D Quadrotor (Fixed Obstacles) | | | | | |
| PPO-Lag | 0.0 | 3 | 85.2 | 81.2 | $132 \pm 11$ |
| PPO-PID | 0.0 | 3 | 89.4 | 88.3 | $\mathbf{145 \pm 12}$ |
| CAP | 0.0 | 4 | 63.6 | 59.2 | $141 \pm 11$ |
| MBPPO | 41.1 | 1 | 75.4 | 73.1 | $132 \pm 9$ |
| CBF-RL | 0.0 | 2 | 82.3 | 79.2 | $140 \pm 10$ |
| RESPO | 0.0 | 1 | 65.7 | 21.3 | $79 \pm 8$ |
| VSRL | **100.0** | **15** | **100.0** | **100.0** | $122 \pm 14$ |

**Lane Following:** Our lane following environment follows the discrete-time bicycle model [Kong et al., 2015]. The model inputs are 3-dimensional $(x, \theta, v)$, where $x$ is the lateral distance to the center of the lane, $\theta$ is the angle relative to the center of the lane, and $v$ represents the speed. The objective is to maintain a constant speed while following the lane, meaning the system equilibrium point is $(x, \theta, v) = (0, 0, v_{\text{target}})$. The safety constraints are 1) $x$ stays within a maximum distance from the lane center ($\|x\| \leq d_{\max}$), 2) $\theta$ remains within a predefined range ($\|\theta\| \leq \theta_{\max}$), and 3) $v$ does not exceed the maximum threshold ($v \leq v_{\max}$).

**Vehicle Avoidance:** Our vehicle avoidance environment features a vehicle moving on an $x$-$y$ plane, with 4-dimensional inputs $(x, y, \theta, v)$. Here, $(x, y)$ represents the location of the vehicle on the plane, $\theta$ is the angle relative to the $y$-axis, and $v$ is the speed. In this setting, we have five moving obstacles, each moving from one point to another at constant speed. Each obstacle is represented as a square. Additionally, safety constraints are set for the speed ($v \leq v_{\max}$) and angle ($\|\theta\| \leq \theta_{\max}$). The task is to navigate the vehicle to a designated location while following safety constraints.

**2D Quadrotor:** For the 2D quadrotor environment, we follow the settings in Dai et al. [2021]. The input is 6-dimensional $(y, z, \theta, \dot{y}, \dot{z}, \dot{\theta})$, where $(y, z)$ represents the position of the quadrotor on the $y$-$z$ plane, and $\theta$ represents the angle. The action space is 2-dimensional and continuous; the actions are clipped within a range to reflect motor constraints. Our safety criteria include an angle constraint ($\|\theta\| \leq \theta_{\max}$) and a minimum height constraint to prevent collision with the ground ($y \geq y_{\min}$). We consider two scenarios for obstacles: fixed and moving. For fixed obstacles, there are five rectangular obstacles positioned in the $y$-$z$ plane. For moving obstacles, there are five obstacles that moves from one point to another at constant speed, each represented as a square.

**3D Quadrotor:** Our 3D quadrotor environment features a 12-dimensional input space, represented as $(x, y, z, \phi, \theta, \psi, \dot{x}, \dot{y}, \dot{z}, \omega_x, \omega_y, \omega_z)$. The action space is 4-dimensional and continuous; the actions are clipped within a range to reflect motor constraints. Here, $(x, y, z)$ denotes the location of the quadrotor in space, $\phi$ is the roll angle, $\theta$ is the pitch angle, and $\psi$ is the yaw angle, $\omega_x, \omega_y, \omega_z$ represent the angular velocity around the $x$, $y$, and $z$ axes, respectively. The task is to navigating towards the goal while adhering to safety constraints, which include avoiding five obstacles represented as 3D rectangles. The details for the environment settings are deferred to the Appendix.

## 4.2 Results

As shown in Table 1, our approach significantly outperforms all baselines in terms of verified safety, as well as empirical safety over the entire episode horizon. Furthermore, the only environment in which VSRL exhibits a significant decrease in reward compared to baselines is lane following; for the rest, it achieves reward comparable to, or better than the baselines.

Specifically, in the *lane following* environment, the proposed VSRL approach achieves verified 80-step safety using a single controller (i.e., $|\Theta| = 1$). This is an order of magnitude higher $K$ than all baselines (which only achieve $K \leq 8$). While all baselines obtain a safety record of over 99.9% over the entire episode ($K = 500$), our approach empirically achieves perfect safety.

For vehicle avoidance, we achieve verified 50-step safety using two controllers (i.e., $|\Theta| = 2$); in contrast, the best baseline yields only $K = 13$. We also observe considerable improvements in both verified and empirical safety over the baseline approaches: for example, the best verified baseline (CAP) violates safety over 10% of the time over the full episode length, whereas VSRL maintains a perfect safety record. In this case, VSRL also achieves the highest reward.

For the 2D Quadrotor environment with fixed and moving obstacles, we are able to achieve verified 50-step safety using four and two controllers, respectively. The best baseline achieves only $K = 14$ in the case of fixed and $K = 8$ in the case of moving obstacles (notably, different baselines are best in these cases).

Finally, in the most complex 3D Quadrotor environment, we achieve verified safety for $K = 15$, but empirically maintain a perfect safety record for the entire episode duruction. The best baseline achieves verified safety for only $K = 4$, but is empirically unsafe over 40% of the time during an episode. Even the best safety record of any baseline is unsafe nearly 12% of the time, and we can only verify its safety over a horizon $K = 3$.

**Ablation Study:** We evaluate the importance of both incremental verification and using multiple initial-state-dependent controllers as part of VSRL. As shown in the Appendix (Section A.1), the

former significantly reduces average verification time during training, whereas the latter enables us to greatly boost the size of the initial state region $\mathcal{S}_0$ for which we are able to achieve verify safety.

## 5    Conclusion

We present an approach for learning neural network control policies for nonlinear neural dynamical systems. In contrast to conventional methods for safe control synthesis which rely on forward invariance-based proofs, we opt instead for the more pragmatic finite-step reachability verification. This enables us to make use of state-of-the-art differentiable neural network overapproximation tools that we combine with three key innovations. The first is a novel curriculum learning approach for maximizing safety horizon. The second is to learn multiple initial-state-dependent controllers. The third is to leverage small changes in iterative gradient-based learning to enable incremental verification. We show that the proposed approach significantly outperforms state of the art safe RL baselines on several dynamical system environments, accounting for both fixed and moving obstacles. A key limitation of our approach is the clearly weaker safety guarantees it provides compared to forward invariance. Nevertheless, our results demonstrate that finite-step reachability provides a more pragmatic way of achieving verified safety that effectively achieves safety over the entire episode horizon *in practice*, providing an alternative direction for advances in verified safe RL to the more typical forward-invariance-based synthesis.

## Acknowledgments

This research was partially supported by the NSF (grants IIS-1905558, IIS-2214141, CCF-2403758, and IIS-2331967) and NVIDIA. Huan Zhang is supported in part by the AI2050 program at Schmidt Sciences (AI 2050 Early Career Fellowship).

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

# A  Appendix

## A.1  Ablation Study

In this section, we conduct an ablation study to evaluate the importance of both incremental verification and the use of multiple initial-state-dependent controllers as part of the VSRL approach.

Table 2: Runtime (in seconds) for 20 training epochs with and without incremental verification.

| | Lane Following | | | |
| | 5-step ($\downarrow$) | 10-step ($\downarrow$) | 15-step ($\downarrow$) | 20-step ($\downarrow$) |
|---|---|---|---|---|
| w/ Incr. Veri. | **9.4** | **9.4** | **17.4** | **24.8** |
| w/o Incr. Veri. | 9.6 | 38.1 | 105.9 | 185.5 |
| | Vehicle Avoidance | | | |
| | 5-step ($\downarrow$) | 10-step ($\downarrow$) | 15-step ($\downarrow$) | 20-step ($\downarrow$) |
| w/ Incr. Veri. | **14.0** | **16.1** | **19.8** | **25.5** |
| w/o Incr. Veri. | 14.1 | 47.6 | 110.5 | 187.1 |
| | 2D Quadrotor | | | |
| | 5-step ($\downarrow$) | 10-step ($\downarrow$) | 15-step ($\downarrow$) | 20-step ($\downarrow$) |
| w/ Incr. Veri. | **8.4** | **13.7** | **15.9** | **19.6** |
| w/o Incr. Veri. | 8.8 | 35.8 | 86.8 | 152.1 |
| | 3D Quadrotor | | | |
| | 5-step ($\downarrow$) | 6-step ($\downarrow$) | 7-step ($\downarrow$) | 8-step ($\downarrow$) |
| w/ Incr. Veri. | 31.0 | **31.9** | **36.7** | **49.9** |
| w/o Incr. Veri. | **30.9** | 61.6 | 149.9 | 403.6 |

Table 2 presents the ablation study results for incremental verification. To ensure a fair comparison, we record the runtime for 20 training epochs with only one region from the grid split for all environments. In practice, this process can be run on GPUs in parallel for multiple regions. Given that the neural network structures for the 2D Quadrotor environment with both moving and fixed obstacles are the same, the runtime results are similar; therefore, we report these collectively as 2D Quadrotor. The results indicate that incremental verification significantly reduces the average verification time during training. Without incremental verification, the verification time increases rapidly as the number of steps increases.

Table 3: Percentage of regions in $\mathcal{S}_0$ that can be verified for safety for $K$ steps (Verified-$K$).

| | Veh. Avoid. ($\uparrow$) | 2D-Quad (F) ($\uparrow$) | 2D-Quad (M) ($\uparrow$) | 3D-Quad (F) ($\uparrow$) |
|---|---|---|---|---|
| Single Ctrl. | 99.0 | 97.6 | 96.9 | 74.7 |
| Multi Ctrl. | **100.0** | **100.0** | **100.0** | **100.0** |

Table 3 shows the ablation study results for using multiple initial-state-dependent controllers. We report results for the Vehicle Avoidance environment (Veh. Avoid.), 2D Quadrotor with fixed obstacles (2D-Quad (F)), moving obstacles (2D-Quad (M)), and 3D Quadrotor (3D-Quad (F)). We exclude the Lane Following environment from this comparison, as only one controller was used there to achieve 100% verified safety. The results demonstrate that using multiple controllers significantly enhances the ability to achieve verified safety across a larger initial state region $\mathcal{S}_0$.

## A.2  Experiment Setup

**Lane Following** Our lane following environment follows the discrete-time bicycle model [Kong et al., 2015]

$$\dot{x} = v\cos(\theta + \beta)$$

$$\dot{y} = v \sin(\theta + \beta)$$
$$\dot{\theta} = \frac{v}{l_r} \sin(\beta)$$
$$\dot{v} = a$$
$$\beta = \tan^{-1}\left(\frac{l_r}{l_f + l_r} \tan(\delta_f)\right)$$

where we set the wheel base of the vehicle to 2.9m. The model inputs are 3-dimensional $(x, \theta, v)$, where $x$ is the lateral distance to the center of the lane, $\theta$ is the angle relative to the center of the lane, and $v$ represents the speed. The objective is to maintain a constant speed while following the lane, meaning the system equilibrium point is $(x, \theta, v) = (0, 0, v_{\text{target}})$. The safety constraints are

1. $x$ stays within a maximum distance from the lane center ($\|x\| \leq d_{\max}$),
2. $\theta$ remains within a predefined range ($\|\theta\| \leq \theta_{\max}$),
3. $v$ does not exceed the maximum threshold ($v \leq v_{\max}$).

The parameters are set as $d_{\max} = 0.7$, $\theta_{\max} = \pi/4$, and $v_{\max} = 5.0$. The initial regions $\mathcal{S}_0$ is $x \in [-0.5, 0.5], \theta \in [-0.2, 0.2], v \in [0.0, 0.5]$. The reward received at each step is measured as the distance to the equilibrium point. More specifically, for a state that is of distance $d$ to the target equilibrium point, the reward is $e^{-d}$. For VSRL training, our controller is initialized using a controller pretrained with a safe RL algorithm. When training with the bound loss, we add a large penalty on unsafe states to incentivize maintaining safety throughout the entire trajectory. For branch and bound during verification, we set the precision limit as $0.025$, which means as soon as the precision of the grid region reaches this precision, we stop branching. For the dynamics approximation, we use an NN with two layers of ReLU each of size 8.

**Vehicle Avoidance**   Our vehicle avoidance environment features a vehicle moving on an $x$-$y$ plane, with 4-dimensional inputs $(x, y, \theta, v)$. Here, $(x, y)$ represents the location of the vehicle on the plane, $\theta$ is the angle relative to the $y$-axis, and $v$ is the speed. In this setting, we have five moving obstacles, each moving from one point to another at a constant speed for the duration of 500 steps. The five obstacles are: 1) moving from $(x, y) = (-0.6, 1.0)$ to $(x, y) = (-0.35, 2.0)$; 2) moving from $(x, y) = (0.6, 0.0)$ to $(x, y) = (0.75, 1.0)$; 3) moving from $(x, y) = (0.0, 1.0)$ to $(x, y) = (0.0, 2.0)$; 4) moving from $(x, y) = (-0.85, 1.0)$ to $(x, y) = (-1.6, 1.5)$; 5) moving from $(x, y) = (0.75, 0.0)$ to $(x, y) = (0.85, 0.0)$. Each obstacle is represented as a square with a diameter of $0.1$. Additionally, safety constraints are set for the speed ($v \leq v_{\max}$) and angle ($\|\theta\| \leq \theta_{\max}$), where $v_{\max} = 5.0$ and $\theta_{\max} = \pi/2$. The task is to navigate the vehicle to a designated location while following safety constraints. The agent starts near the origin within an area defined by $x, y \in [-0.5, 0.5]$, $\theta \in [-0.2, 0.2]$, and $v \in [0, 0.1]$, and the goal is $(x_{\text{target}}, y_{\text{target}}) = (1.0, 2.0)$. The branching precision limit is $0.025$ and for dynamics approximation, we use a NN with two layers of ReLU each of size 10.

**2D Quadrotor**   For the 2D quadrotor environment, we follow the settings in Dai et al. [2021].
$$\dot{x} = -\frac{\sin(\theta)}{m} \cdot (u_0 + u_1)$$
$$\dot{y} = \frac{\cos(\theta)}{m} \cdot (u_0 + u_1) - g$$
$$\dot{\theta} = \frac{\text{length}}{I} \cdot (u_0 - u_1)$$

We use a timestep $dt = 0.02$, the mass of the quadrotor is set to $m = 0.486$, the length to $l = 0.25$, the inertia to $I = 0.00383$, and gravity to $g = 9.81$. The input is 6-dimensional $(y, z, \theta, \dot{y}, \dot{z}, \dot{\theta})$, where $(y, z)$ represents the position of the quadrotor on the $y$-$z$ plane, and $\theta$ represents the angle. The action space is 2-dimensional and continuous; the actions are clipped within a range to reflect motor constraints. Our safety criteria are

1. angle $\theta$ remains within a predefined range ($\|\theta\| \leq \theta_{\max}$),
2. a minimum height constraint to prevent collision with the ground ($y \geq y_{\min}$),
3. avoid obstacles.

Here we set $\theta_{\max} = \pi/3$ and $y_{\min} = -0.2$. The task is for the quadrotor to navigate towards the goal while following safety constraints. We consider two scenarios for obstacles: fixed and moving. For

fixed obstacles, there are five rectangular obstacles positioned in the $y$-$z$ plane. We use $(x_l, x_u, y_l, y_u)$ to represent the two dimensional box, and the obstacles are: $(x_l, x_u, y_l, y_u) = (-0.3, -0.1, 0.4, 0.6)$, $(x_l, x_u, y_l, y_u) = (-1.2, -0.8, 0.2, 0.4)$, $(x_l, x_u, y_l, y_u) = (0.0, 0.1, 0.5, 1.0)$, $(x_l, x_u, y_l, y_u) = (0.6, 0.7, 0.0, 0.2)$, $(x_l, x_u, y_l, y_u) = (-0.8, -0.7, 0.7, 0.9)$. For moving obstacles, there are five obstacles that moves from one point to another at constant speed for the duration of 500 steps, each represented as a square of diameter 0.1. The obstacles are: 1) moving from $(x, y) = (0.6, 0.0)$ to $(x, y) = (0.6, 0.1)$; 2) moving from $(x, y) = (-0.5, 0.2)$ to $(x, y) = (-0.4, 0.3)$; 3) moving from $(x, y) = (-0.3, 0.4)$ to $(x, y) = (-0.4, 0.5)$; 4) moving from $(x, y) = (-0.1, 0.3)$ to $(x, y) = (0.0, 0.4)$; 5) moving from $(x, y) = (-0.7, 0.5)$ to $(x, y) = (-0.4, 0.6)$. The initial region for the quadrotor is defined with $x \in [-0.5, 0.5]$ and the remaining state variables within $[-0.1, 0.1]$. The target goal is set to $(x, y) = (0.6, 0.6)$. We set the branching limit to $0.0125$ and for dynamics approximation we use a NN with two layers of ReLU each of size 6.

**3D Quadrotor**    Our 3D quadrotor environment features a 12-dimensional input space, represented as $(x, y, z, \phi, \theta, \psi, \dot{x}, \dot{y}, \dot{z}, \omega_x, \omega_y, \omega_z)$. The action space is 4-dimensional and continuous; the actions are clipped within a range to reflect motor constraints. Here, $(x, y, z)$ denotes the location of the quadrotor in space, $\phi$ is the roll angle, $\theta$ is the pitch angle, and $\psi$ is the yaw angle, $\omega_x, \omega_y, \omega_z$ represent the angular velocity around the $x$, $y$, and $z$ axes, respectively. The environment setting and neural network dynamics approximation follows the setup in Dai et al. [2021], with the modification of using ReLU activations instead of LeakyReLU. The system dynamics is:

$$\text{plant\_input} = \begin{bmatrix} 1 & 1 & 1 & 1 \\ 0 & L & 0 & -L \\ -L & 0 & L & 0 \\ \kappa_z & -\kappa_z & \kappa_z & -\kappa_z \end{bmatrix} \cdot u$$

$$R = \text{rpy2rotmat}(\phi, \theta, \psi)$$

$$\ddot{\mathbf{p}} = \begin{bmatrix} 0 \\ 0 \\ -g \end{bmatrix} + R \cdot \begin{bmatrix} 0 \\ 0 \\ \text{plant\_input}[0]/m \end{bmatrix}$$

$$\dot{\boldsymbol{\omega}} = \frac{-\boldsymbol{\omega} \times (I \cdot \boldsymbol{\omega}) + \text{plant\_input}[1 :]}{I}$$

$$\begin{bmatrix} \dot{\phi} \\ \dot{\theta} \\ \dot{\psi} \end{bmatrix} = \begin{bmatrix} 1 & \sin(\phi) \cdot \tan(\theta) & \cos(\phi) \cdot \tan(\theta) \\ 0 & \cos(\phi) & -\sin(\phi) \\ 0 & \frac{\sin(\phi)}{\cos(\theta)} & \frac{\cos(\phi)}{\cos(\theta)} \end{bmatrix} \cdot \boldsymbol{\omega}$$

The dynamics neural network has two ReLU layers, each with a size of 16 and $dt = 0.02$. We set the branching precision limit to $0.00625$. The task is to navigating towards the goal while avoiding five obstacles represented as 3D rectangles. The locations of the obstacles are $(-0.5, 0.5, -0.2, 0.2, -0.65, -0.55), (-0.7, -0.6, -0.1, 0.1, -0.5, -0.4), (0.5, 0.6, -0.2, 0.2, -0.4, -0.3), (-0.8, -0.6, 0.2, 0.4, -0.3, -0.2), (-0.8, -0.6, -0.4, -0.2, -0.2, -0.1)$, where the first obstacle is to avoid controller collide with the ground. We set the goal at $(x, y, z) = (0.0, 0.0, 0.0)$, and the initial region is defined with $x \in [-0.5, 0.5]$, $y \in [-0.1, 0.1]$, and $z \in [-0.5, -0.3]$, with the remaining variables confined to the range $[-0.05, 0.05]$. The reward is calculated based on the distance to the goal, where the agent receives a higher reward for being closer to the goal. The environment episodes end if either the magnitude of $\phi$ or $\theta$ exceeds $\pi/3$.

### A.3   Compute Resources

Our code runs on an AMD Ryzen 9 5900X CPU with a 12-core processor and an NVIDIA GeForce RTX 3090 GPU.

