# OpenReview forum: "Verified Safe Reinforcement Learning  for Neural Network Dynamic Models"
_NeurIPS.cc/2024/Conference — NeurIPS 2024 poster_

### Official Review · Reviewer_8TUT · 2024-07-11

**Soundness:** 3
**Presentation:** 4
**Contribution:** 3
**Rating:** 7
**Confidence:** 4

**Summary:**

The paper proposes methods to learn formally verified neural network control policies for (continuous-space, discrete-time) non-linear dynamical systems, whose dynamics are also represented by a neural network. It builds on existing methods for NN verification, and their application to a k-step composition of NNs for the policy and dynamics. The key novel ideas of the paper are a variant of curriculum learning, built into an incremental policy synthesis approach over increasing time horizons, and a parameterized approach to representing state-dependent policies. Experimental results show improved performance on four benchmarks compared to five comparable techniques.

**Strengths:**

- Clear and very well written paper.

- Clearly presented and motivated novel ideas to improve performance on a challenging problem.

- Comprehensive empirical evaluation with a good number of meaningful benchmarks and baselines, and a further ablation study in the appendix.

- Impressive gains over baseline implementations.

**Weaknesses:**

- The empirical results focus on the degree of safety achieved by different policies, but there seems to be no discussion of performance, e.g. runtime, or more directly the scalability of the various methods.

**Questions:**

1. Relating to the stated weakness above, what is the experimental setup in terms of timeout (if any) used to compare tools, and what is the limiting factor in terms of the techniques' ability to synthesise safe policies?

**Limitations:**

Yes

---

> ### Author Rebuttal · Authors · 2024-08-06
>
> We appreciate the reviewer's thoughtful comments and feedback. Our responses are below.
>
> > **Question 1:** What is the experimental setup in terms of timeout (if any) used to compare tools.
>
> **Response 1:**
> We did not use any timeouts in our experiments, as all algorithms completed within reasonable time. Table 2 (in the Appendix) provides some timing details as an ablation study for our approach. We will add complete details in the revision.
>
> > **Question 2:** What is the limiting factor in terms of the techniques' ability to synthesise safe policies?
>
> **Response 2:**
> The primary limiting factor is the scailablity of the verification tool for more complex domains, particularly when we include moving obstacles. This is of two types: 1) actual verification (e.g., after training is completed), and 2) differentiable verification that training requires. The latter is a more significant limiting factor than the former, and our approach would directly benefit from further advances in this area.

---

> > ### Comment · Reviewer_8TUT · 2024-08-11
> >
> > Thank you for these clarifications.

---

### Official Review · Reviewer_gxvV · 2024-07-11

**Soundness:** 2
**Presentation:** 3
**Contribution:** 2
**Rating:** 6
**Confidence:** 4

**Summary:**

The authors primarily propose a novel method to learn verified safe control policies for nonlinear neural dynamical systems, aiming to achieve safety in the sense of bounded reachability. By leveraging memorization, forward reachability analysis, and differentiable reachability over-approximation, the authors effectively learn verified safe policies, and further introduce an incremental verification approach to enhance the efficiency of the learning process, enabling the acquisition of multiple verified initial state-dependent controllers.

**Strengths:**

The manuscript proposes a method for learning a k-step verified safe neural network controller, aimed at maximizing the efficiency of systems with neural dynamics. In contrast to traditional approaches relying on forward-invariance proofs for safe control synthesis, the manuscript opts for the more practical bounded reachability verification, enabling the leveraging of state-of-the-art differentiable neural network approximation tools.

The experimental results demonstrate that, across several dynamic system environments considering both static and moving obstacles, the proposed approach significantly outperforms the state-of-the-art safe reinforcement learning baselines. This highlights the advantages of the bounded reachability verification framework over the traditional forward-invariance guarantees, in terms of practical viability and performance when applied to systems with neural dynamics.

**Weaknesses:**

The algorithm proposed in this manuscript lacks rationality analysis

(1) Page 3.: “and required statistical assumptions“->”and requires statistical assumptions”

(2) Page 4.: “which enable learning verified safe controllers over longer horizons K”->”which enables learning verified safe controllers over longer horizons K”

(3) Page 4.: “In this work, we primarily utilize the α,β-CROWN toolbox”-> The α,β-CROWN toolbox should give a brief introduction here

**Questions:**

(1) Can the author explain the soundness of the approach proposed in the manuscript?

(2) The rationality of the experimental comparison baseline, why is it more interesting to compare with these methods? What are the maximum dimensions of the system security verification problem that can be solved by this method?

**Limitations:**

The manuscript thoroughly discusses the limitations of the proposed methods, and the security guarantees provided are weaker compared to forward-invariance.  However, bounded reachability offers a more practical approach to achieving verified safety, effectively realizing the safety of the entire event horizon in practice, and provides an alternative for the development of verified safe reinforcement learning.

---

> ### Author Rebuttal · Authors · 2024-08-06
>
> We thank the reviewer for the thoughtful comments and suggestions. Our responses are below.
>
> > **Question 1:** Can the author explain the soundness of the approach proposed in the manuscript?
>
> **Response 1:**
> Soundness is a direct consequence of our use of a sound verification tool $(\alpha,\beta)$-CROWN, before declaring any of our initial-state-dependent controllers safe in the sense of $K$-step reachability (Lines 4-5 and 10-11 in Algorithm 2). We will clarify in the revision.
>
> > **Question 2:** The rationality of the experimental comparison baseline, why is it more interesting to compare with these methods?
>
> **Response 2:**
> We chose recent baselines that cover the classical and SOTA approaches for a wide range of methods commonly used in Safe RL, including Lagrangian penalty, reward penalty, constrained PPO, and safe RL with reachability estimation. We also added a control-theoretic baseline using CBF in response to Reviewer kLbx (see Response 2 there); the performance of this baseline is comparable to our other baselines.
>
> > **Question 3:** What are the maximum dimensions of the system security verification problem that can be solved by this method?
>
> **Response 3:**
> This depends greatly on the horizon $K$ that we wish to verify. For example, for small reachability horizon $K$, we can scale to thousands of dimensions. However, as meaningful reachability horizon entails larger $K$, this limits scalability, with verification tools the primary bottleneck. In general, there will be a tradeoff between state space dimension, verifiable horizon $K$, and fraction of state space that we can prove safety for.

---

> > ### Comment · Reviewer_gxvV · 2024-08-10
> >
> > Thank you for your response. The authors have addressed my main concern with the paper. As such, I've increased the score to Weak Accept.

---

### Official Review · Reviewer_11TB · 2024-07-11

**Soundness:** 3
**Presentation:** 3
**Contribution:** 3
**Rating:** 6
**Confidence:** 4

**Summary:**

This paper tackles the problem of synthesizing verified control policies for dynamical systems with the use of neural networks. The authors focus on nonlinear discrete-time system dynamics for which the use of neural networks is motivated by the challenge of reaching the goal without colliding with obstacles. To guarantee non-collision and maintain reasonable performance, the authors propose an approach of iteratively using an existing verification toolbox with curriculum-based lookahead. The approach is evaluated on the benchmarks from previous work.

**Strengths:**

The approach of building a curriculum of horizons and re-using verification results from the previous steps is relatively novel and provides an interesting future direction.

It is a significant improvement over the baselines that the proposed algorithm achieves 100% empirical safety. The decrease in performance is evident for several benchmarks and may exacerbate for a longer horizon.

The paper is overall well-structured. However, a motivating example to guide the reader through the algorithm would be helpful.

**Weaknesses:**

As can be seen from the ablation study, the runtime improvement is achieved with the proposed approach when the lookahead is longer than 5 steps. It is not clear if a long lookahead is required for the considered benchmarks.
The authors motivate the choice of verification toolbox (a,b-CROWN) with its suitability for incremental verification. It is not clear how much the proposed technique is constrained by the chosen toolbox or general enough to be adopted for other verification tools under certain assumptions.

The choice of benchmark problems is not motivated in the evaluation section. The description of the benchmarks is not complete and requires looking into original papers. It would be appreciated to have complete system models stated formally in the appendix.

**Questions:**

1. Could the authors explain the choice of these particular benchmark problems to evaluate the approach on?
2. Is the state-space continuous or discrete for these benchmarks? (it appears it is continuous but only action-space is explicitly mentioned)
3. Which of the chosen benchmark problems particularly showcases the benefit of a longer-than-five step lookahead reachability?
4. How is the average reward computed (with verification over what horizon size)? Is it dependent on the verification horizon?
5. Is it possible to incorporate the proposed forward K-step reachability into the related reachability-based verification tools which do not yet consider it? What features or assumptions need to be satisfied?

**Limitations:**

The approach seems to have been specifically designed for verification tools that are suitable for incorporating into gradient-based learning. This can be a significant limitation to generalizability of the approach, since the authors do not discuss or evaluate what requirements the verification tools need to satisfy to be adopt the proposed approach. Based on the current presentation, the approach can only work with a,b-CROWN toolbox. If this is the case, it must be explicitly discussed as a limitation, it is, however, still an improvement over this particular domain.

Open-access to data and code: the answer is "yes" when the data and code are anonymously open-source. If they are not yet open-sourced the answer should be "no yet".

---

> ### Author Rebuttal · Authors · 2024-08-06
>
> We thank the reviewer for the detailed comments and feedback. Our responses are below.
>
> > **Question 1:** A motivating example to guide the reader through the algorithm would be helpful.
>
> **Response 1:**
> We appreciate the suggestion and will use vehicle avoidance as such an example in the revision.
>
> > **Question 2:** As can be seen from the ablation study, the runtime improvement is achieved with the proposed approach when the lookahead is longer than 5 steps. It is not clear if a long lookahead is required for the considered benchmarks.
>
> **Response 2:**
> Below is the result where we only train for 5 steps lookahead.
> |                               |  Verified-K  |  Verified-Max  |  Emp-K |  Emp-500  |  Avg   Reward   |
> |-------------------------------|--------------|----------------|--------|-----------|-----------------|
> | Lane Following, K = 80        | 98.7         | 7              | 99.9   | 99.9      |  328 $\pm$ 5    |
> | Vehicle Avoidance (M), K = 50 | 73.1         | 6              | 88.7   | 88.7      |  304 $\pm$ 10   |
> | 2D Quadrotor (F), K = 50      | 0.0            | 5              | 87.5   | 87.5      |  401 $\pm$ 23   |
> | 2D Quadrotor (M), K = 50      | 0.0            | 5              | 99.7   | 99.7      |  373 $\pm$ 5    |
> | 3D Quadrotor (F), K = 15      | 0.0            | 5              | 90.5   | 89.3      |  129 $\pm$ 10   |
>
> This ablation (in comparison with the results in Table 1 in the paper) shows why considering lookahead with $K>5$ is critical for our benchmarks (especially 2D and 3D Quadrotor).
>
> > **Question 3:** It is not clear how much the proposed technique is constrained by the chosen toolbox or general enough to be adopted for other verification tools under certain assumptions.
>
> **Response 3:**
> Our training framework is general and can in principle work with any *differentiable* verification technique; to our knowledge $(\alpha,\beta)$-CROWN is simply the best differentiable neural network verifier [5]. We will clarify in the revision.
>
> > **Question 4:** The choice of benchmark problems is not motivated in the evaluation section. The description of the benchmarks is not complete and requires looking into original papers. It would be appreciated to have complete system models stated formally in the appendix.
>
> **Response 4:**
> Our benchmark problems are common benchmarks in the literature. For example, [3,4] use Quad2D, and [4] additionally uses Quad3D. The vehicle avoidance benchmark is a variant of CarGoal in the Safety Gym, used by PPO-PID, MBPPO, and RESPO baselines. We will add the description of the complete system models in the final version of the paper.
>
> > **Question 5:** Is the state-space continuous or discrete for these benchmarks?
>
> **Response 5:**
> Both action and state space are continuous for all.
>
> >**Question 6:** How is the average reward computed? Is it dependent on the verification horizon?
>
> **Response 6:**
> Average reward is computed over the entire episode horizon for each environnment, independently of the verification horizon (just as it is done in conventional RL). We will clarify in the revision.
>
> > **Question 7:** Is it possible to incorporate the proposed forward K-step reachability into the related reachability-based verification tools which do not yet consider it? What features or assumptions need to be satisfied?
>
> **Response 7:**
> Yes. For training, the reachability-based verification toolbox needs to be differentiable. For verification, any reachability tool that supports NN-based dynamics and controllers would work.
>
> > **Question 8:** Open-access to data and code: the answer is "yes" when the data and code are anonymously open-source. If they are not yet open-sourced the answer should be "no yet".
>
> **Response 8:**
> This is a good point; we do intend to make all code and data available on github.
>
> [3] Emam, Yousef, Gennaro Notomista, Paul Glotfelter, Zsolt Kira, and Magnus Egerstedt. "Safe reinforcement learning using robust control barrier functions." IEEE Robotics and Automation Letters, 2022.
>
> [4] Dawson, Charles, Zengyi Qin, Sicun Gao, and Chuchu Fan. "Safe nonlinear control using robust neural lyapunov-barrier functions." In Conference on Robot Learning, 2022.
>
> [5] Brix, Christopher, Stanley Bak, Changliu Liu, and Taylor T. Johnson. "The fourth international verification of neural networks competition (VNN-COMP 2023): Summary and results.", 2023.

---

> > ### Comment · Reviewer_11TB · 2024-08-09
> >
> > Thank you for your detailed response and promise to clarify the points in the revision. I have no further questions.

---

### Official Review · Reviewer_kLbx · 2024-07-11

**Soundness:** 3
**Presentation:** 3
**Contribution:** 3
**Rating:** 7
**Confidence:** 4

**Summary:**

This paper studies safe reinforcement/control learning by optimization of long-horizon safety verification and learning multiple initial-state-dependent controllers. The authors propose several novel ideas, including a curriculum learning to increase the verification horizon, incremental verification, and split the initial region for multi-controllers training. A set of control tasks show the effectiveness of the approach, compared to other CMDP-based safe RL methods.

**Strengths:**

1. The paper is well-written and easy to follow.
2. As far as I can tell, the idea of optimizing the verification horizon (curriculum learning) is novel and the experimental result is quite significant.
3. The approach is sound.

**Weaknesses:**

1. the reviewer feels that incremental verification in this paper is quite common in the reachable set computation of neural network-controlled systems (NNCS), where the computed reachable set at $k$-th step will become the initial set for $k+1$ set computation. The authors may have to discuss their incremental verification vs. NNCS verification tools. To name a few tools, please refer to POLAR-Express[1], CORA[2], etc.
2. Splitting the state/initial space set into small grids for verification and controller synthesis/training is also common in the NNCS verification community. However, I have to admit that the design of multi-initial-state-dependent controllers is new to me.
3. The experiments only compare to the CMDP-based approach, it is unclear how the proposed approach compares to other control-theoretical methods, for instance, the CBF-based approaches.

[1] Wang, Y., Zhou, W., Fan, J., Wang, Z., Li, J., Chen, X., ... & Zhu, Q. (2023). Polar-express: Efficient and precise formal reachability analysis of neural-network controlled systems. IEEE Transactions on Computer-Aided Design of Integrated Circuits and Systems. https://github.com/ChaoHuang2018/POLAR_Tool
[2] https://github.com/TUMcps/CORA?tab=readme-ov-file

**Questions:**

1. How to ensure that Algorithm 2 can terminate?

**Limitations:**

The authors have addressed the limitations in the final part.

---

> ### Author Rebuttal · Authors · 2024-08-06
>
> We thank the reviewer for the thoughtful comments and suggestions. Our responses are below.
>
>  > **Question 1:** The authors may have to discuss their incremental verification vs. NNCS verification tools; POLAR-Express[1], CORA[2], etc.
>
> **Response 1:** Indeed, incremental verification is a well-explored idea in the verification literature, and we took inspiration from this literature. To our knowledge, we are the first to use this idea in *training* provably safe controllers. This entails several innovations:
> 1) We incrementally verify (and backpropagate the results) several steps ahead in a single training iteration (i.e., not merely from $k$ to $k+1$, but more generally from $k_i$ to $k_{i+1}$, where $k_{i+1} - k_i > 1$). Doing this more generalized version of incremental verification is crucial for training, significantly speeding it up, and reducing the likelihood of being stuck in "local optima" where inertia resulting from policy obtained for $k$ prevents verification from succeeding for $k+1$ (e.g., because we are too close to the unsafe region with velocity directed towards it).
> 2) Incremental verification is also an important component of our  "Curriculum Learning with Memorization" training piece, where a significant challenge is backpropagation of $\mathcal{L}_{\text{bound}}$, which becomes GPU-intensive as the number of forward reachability steps, and consequently the neural network depth, increases.  (GPU requirements for $K$-step backpropagation significantly exceed those for $K$-step verification.)  Using incremental verifiation also allow us to efficiently extract useful gradient information for training. Furthermore, as shown in the ablation study in Table 2, the use of incremental verification during training significantly increases computational efficiency.
>
> We will add the discussion regarding how our approach to incremental verification compares to, and differs from NNCS verification (e.g., POLAR-Express[1], CORA[2], etc.) in the revision.
>
> >**Question 2:** It is unclear how the proposed approach compares to other control-theoretical methods, for instance, the CBF-based approaches.
>
> **Response 2:**
> We thank the reviewer for the suggestion. We ran an experiment with a CBF-based safe RL baseline from [3]. The results are below (where (M) stands for moving obstacles and (F) stands for fixed obstacles), which we will add to the revision, and are qualitatively similar to other baselines we consider.
>
>
> |  |  |  Verified-K  |  Verified-Max  |  Emp-K |  Emp-500  |  Avg   Reward  |
> |---|---|---|---|---|---|---|
> | Lane Following, K = 80 | CBF-based | 98.7 | 7 | 99.9 | 99.9 | $\mathbf{331 \pm 7}$ |
> |  | VSRL (ours) | $\mathbf{100.0}$  |      $\mathbf{80}$        |       $\mathbf{100.0}$      |   $\mathbf{100.0}$        |         $214 \pm 5$  |
> |  |  |  |  |  |  |  |
> | Vehicle Avoidance (M), K = 50 | CBF-based | 73.0 | 6 | 89.3 | 89.3 | $301 \pm 15$ |
> |  | VSRL (ours) | $\mathbf{100.0}$  |      $\mathbf{50}$       |     $\mathbf{100.0}$      |    $\mathbf{100.0}$       |      $\mathbf{401 \pm 4}$    |
> |  |  |  |  |  |  |  |
> | 2D Quadrotor (F), K = 50 | CBF-based | 0.0 | 5 | 89.9 | 89.7 | $\mathbf{408 \pm 17}$ |
> |  | VSRL (ours) |  $\mathbf{100.0}$  |  $\mathbf{50}$  |  $\mathbf{100.0}$  |  $\mathbf{100.0}$ |  ${401 \pm 20}$ |
> |  |  |  |  |  |  |  |
> | 2D Quadrotor (M), K = 50 | CBF-based | 0.0 | 4 | 99.3 | 99.3 | $\mathbf{369 \pm 6}$ |
> |  | VSRL (ours) | $\mathbf{100.0}$  |      $\mathbf{50}$        |     $\mathbf{100.0}$      |    $\mathbf{100.0}$        |    $364\pm 4$   |
> |  |  |  |  |  |  |  |
> | 3D Quadrotor (F), K = 15 | CBF-based | 0.0 | 2 | 82.3 | 79.2 | $\mathbf{140 \pm 10}$ |
> |  | VSRL (ours) | $\mathbf{100.0}$  |     $\mathbf{15}$         |     $\mathbf{100.0}$      |     $\mathbf{100.0}$      |     $122 \pm 14$  |
> |  |  |  |  |  |  |  |
>
> [3] Emam, Yousef, Gennaro Notomista, Paul Glotfelter, Zsolt Kira, and Magnus Egerstedt. "Safe reinforcement learning using robust control barrier functions." IEEE Robotics and Automation Letters, 2022.
>
> > **Question 3: How to ensure that Algorithm 2 can terminate?**
>
> **Response 3:**
> Ensuring termination entails simply setting a limit on the number of training iterations or setting a timeout, as is typically done in RL, or imposing an upper bound on the number of controllers the Algorithm can return. In our experiments, however, this proved unnecessary, as all cases terminated in reasonable time and with few initial-state-dependent controllers.

---

> > ### Comment · Reviewer_kLbx · 2024-08-10
> >
> > I greatly appreciate the feedback from the authors, which has sufficiently addressed my concerns and problems. I am willing to increase my score.

---

### Decision · Program_Chairs · 2024-09-25

**Decision:**

Accept (poster)

**Comment:**

The paper introduces a novel technique for safe RL leveraging differentiable
NN verification tools, currently, the algorithm is built using the $\alpha,\beta$-crown
verification method.

The reviewers agreed that the approach is sound and novel and the authors addressed
the main concerns of the reviewers during the rebuttal phase. In the final version of the paper, please include
the discussion on the neural network-controlled systems, the additional benchmark, ablation and
clarifications provided in the response to the reviewers.